# Characteristics and practices of school-based cluster randomised controlled trials for improving health outcomes in pupils in the UK: a systematic review protocol

Kitty Parker ![ORCID] ,[1] Michael P Nunns,[2] ZhiMin Xiao,[3] Tamsin Ford,[4] Obioha C Ukoumunne ![ORCID] [1]

[1]NIHR ARC South West Peninsula (PenARC), University of Exeter, Exeter, UK
[2]College of Medicine and Health, University of Exeter, Exeter, UK
[3]Graduate School of Education, University of Exeter, Exeter, UK
[4]Department of Psychiatry, University of Cambridge, Cambridge, UK

**Correspondence to**
Kitty Parker;
kp477@exeter.ac.uk

## ABSTRACT

**Introduction** Cluster randomised trials (CRTs) are studies in which groups (clusters) of participants rather than the individuals themselves are randomised to trial arms. CRTs are becoming increasingly relevant for evaluating interventions delivered in school settings for improving the health of children. Schools are a convenient setting for health interventions targeted at children and the CRT design respects the clustered structure in schools (ie, pupils within classrooms/teachers within schools). Some of the methodological challenges of CRTs, such as ethical considerations for enrolment of children into trials and how best to handle the analysis of data from participants (pupils) that change clusters (schools), may be more salient for the school setting. A better understanding of the characteristics and methodological considerations of school-based CRTs of health interventions would inform the design of future similar studies. To our knowledge, this is the only systematic review to focus specifically on the characteristics and methodological practices of CRTs delivered in schools to evaluate interventions for improving health outcomes in pupils in the UK.

**Methods and analysis** We will search for CRTs published from inception to 30 June 2020 inclusively indexed in MEDLINE (Ovid). We will identify relevant articles through title and abstract screening, and subsequent full-text screening for eligibility against predefined inclusion criteria. Disagreements will be resolved through discussion. Two independent reviewers will extract data for each study using a prepiloted data extraction form. Findings will be summarised using descriptive statistics and graphs.

**Ethics and dissemination** This methodological systematic review does not require ethical approval as only secondary data extracted from papers will be analysed and the data are not linked to individual participants. After completion of the systematic review, the data will be analysed, and the findings disseminated through peer-reviewed publications and scientific meetings.

**PROSPERO registration number** CRD42020201792.

## Strengths and limitations of this study

► To our knowledge, this is the first systematic review to describe the characteristics and methodological practices of school-based cluster randomised trials (CRTs) of health interventions in the UK.

► The review has a defined search strategy that is tailored to identifying school-based CRTs, eligibility criteria, and prepiloted screening and data extraction strategies to minimise inaccuracies.

► Two independent reviewers will perform screening and data extraction, with any uncertainty resolved by consulting a third reviewer.

► The review will focus on studies conducted in the UK in order to align with available resources and create a relevant and focused review.

► There is the possibility that we are missing the opportunity to learn from studies in countries that have a similar education system to the UK.

## INTRODUCTION

Cluster randomised trials (CRTs), also known as group randomised or place randomised trials, are studies in which groups (clusters) of participants (eg, general practices, organisations, areas, etc) are randomly allocated to the trial arms, rather than the individual participants on whom outcomes are measured.[1] These studies are in contrast to the more traditional individually randomised trials, where the participants themselves are randomised. The CRT design is commonly used in healthcare research when interventions must be delivered at the cluster level and to minimise contamination of the trial arms that might otherwise occur when individuals are randomised.[2]

A characteristic feature of CRTs is that observations on participants who are in the same cluster are usually more similar than

observations on participants who are from different clusters.[2] For example, patients registered with the same general practice are more likely to have similar health outcomes than those registered with different practices.[3] This similarity, or lack of statistical independence, between observations from the same cluster means that the usual procedures for calculating sample size and analysing data in individually randomised trials should not be used in CRTs.[1] The use of standard sample size methods are likely to result in studies that lack power to detect the specified intervention effect and the use of standard analytical methods may produce results that exaggerate evidence for the true effect of the intervention. Therefore, alternative methods should be used when conducting CRTs.

The intracluster correlation coefficient (ICC), denoted $\rho$, quantifies the similarity of observations of individuals within the same cluster. The ICC is the proportion of the total variability in the trial outcome that is between clusters ($\sigma_b^2$) as opposed to between individuals within clusters ($\sigma_w^2$)[4]:

$$\rho = \frac{\sigma_b^2}{\sigma_b^2 + \sigma_w^2}$$

$\rho$ can take values between 0 and 1. The larger $\rho$ is, the greater the similarity between individuals within clusters, or equivalently the greater variability between clusters.

Information about $\rho$ for the primary outcome is invaluable when designing a CRT. It can be estimated from previous studies or feasibility studies, with a similar cluster structure and outcome to the planned trial.[5 6] Authors of CRTs should report estimates of their ICCs, ideally with confidence intervals (CIs) because they are usually based on studies with relatively few clusters,[7] to aid the design of future similar studies.

When calculating the sample size in CRTs one needs to determine the total number of clusters that need to be recruited and the number of individuals that need to be recruited from within each cluster. Methods for calculating the required sample size need to take the ICC into account. When the number of participants in each cluster is fixed and known in advance, the total number of individuals required in a CRT is calculated by inflating the number of individuals that would be required in an individually randomised trial by the *design effect* (*DE*), which is function of $\rho$:

$$DE = 1 + (n - 1)\rho$$

where $n$ is the number of participants providing outcome data in each cluster (cluster size).[8] Having calculated the total number of participants required, this is divided by the cluster size to obtain the total number of clusters that is required. For scenarios where the total number of clusters available for a trial is known, an alternative calculation based on the same approach is used to calculate the number of participants that need to be recruited from each cluster.[3]

When estimating the intervention effect in CRTs, analytical methods that take account of $\rho$ should be used,[1] otherwise CIs will be too narrow and p values will be too small, resulting in an exaggeration of the amount of evidence for a true intervention effect.[9] Furthermore, the degrees of freedom (df) used for calculating the CI and p value for the intervention effect should take account of the number of clusters.[10 11] A distinction can be made between statistical analyses that are carried out at the cluster level and those that are carried out at the individual level.[8] For cluster-level analyses, the outcome is summarised for each cluster, for example, by calculating the mean for continuous outcomes or percentages for binary outcomes across individuals in the cluster. Standard analytical methods are then used to compare the outcome between the trial arms using the cluster-level summary statistics as the observations. This method of analysis is valid because the cluster is both the unit of randomisation and the unit of analysis.[2] Alternatively, analyses of individual-level data involve the application of statistical methods that allow for the within cluster correlation.[2] This approach is exemplified by methods such as mixed effects ('multilevel') models and marginal models estimated using generalised estimating equations.

CRTs are increasingly used to evaluate interventions for improving health outcomes in children.[12 13] Because of the amount of time children spend in school, it provides a natural setting in which interventions for preventing health problems can be delivered, participating children can be recruited, and health outcomes measured.[13] At a policy level, there is increasing awareness of the potential for using the school setting to deliver, non-pharmacological, complex, prevention public health interventions.[12 14 15] Cluster randomisation is a more natural approach than individual randomisation in the school-based setting. It is often difficult to randomise individuals as pupils belong to predefined clusters (eg, class, year group, school), and contamination between the trial arms can result as pupils interact within clusters. The CRT design respects the clustered structure in schools.

School-based CRTs share the same challenges of trials where other types of cluster are randomised. Within-cluster correlation is expected in school-based CRTs for a number of reasons: parents choose the schools their children attend and this may be related to factors associated with pupil outcomes; the school environment and culture will have a common influence on the pupils; pupils interact within schools and this can result in similar behaviours and outcomes.[12 13 16] Other recognised challenges may be even more salient for trials that randomise schools. There are additional ethical considerations for enrolment of children into trials to ensure pupils remain protected as research subjects.[12] Consent needs to be sought from several key agents including parents, pupils, head teachers and teachers. Consent for the school to be allocated the intervention is usually provided by the head teacher, but there may be interventions delivered to entire classes that some parents do not want their children to receive (eg, aspects of sex education programmes that are not part of the standard curriculum). Retention

of recruited pupils is an issue for trials that have a long follow-up duration and there is the need to consider how best to handle the analysis of data from pupils that change schools (clusters) during the course of the study.[17]

Several books have been published regarding CRT methodology.[1 2 18–20] In addition, there have been a number of reviews of the conduct and reporting quality of CRTs.[12 21–26] One systematic review examined the characteristics and quality of reporting of CRTs worldwide involving children[12] and highlighted the specific difficulties of conducting such studies; nearly three-quarters of the included studies randomised schools as the clusters. That review and our initial scoping research suggests a sharp increase in the number of these studies. No systematic review has specifically focused on the characteristics of CRTs of health interventions delivered in the school-based setting in the UK. Such a review would: provide a pool of relevant knowledge for researchers planning future similar trials in the UK; highlight good practices and common methodological challenges; obtain useful trial-based data on the intracluster (intraschool) correlation coefficient; provide relevant parameter values for simulation-based studies that use synthetic data to assess the statistical properties of methods used to analyse data from school-based CRTs.

This review aims to summarise the characteristics of, and methodological practices in, school-based CRTs with pupil health outcomes in the UK. The review examines several areas, including: participant characteristics; intervention type; recruitment, sampling and allocation methods; consent and ethical approval procedures; retention and analysis methods. The main outcome is a description of the methodological characteristics of school-based CRTs in the UK with a health outcome. Knowledge of the study characteristics and practices of researchers will greatly aid the design of CRTs in the school health setting.

## METHODS

The systematic review will describe the characteristics of CRTs with health outcomes in the school setting. This section contains a description of the methodological strategy based on guidelines from the Preferred Reporting Items for Systematic Reviews and Meta-Analyses (PRISMA) statement.[27]

### Search strategy

Peer reviewed articles written in English, published from inception to 30 June 2020 inclusively indexed in MEDLINE (Ovid) will be the source of data for this systematic review. The search strategy was developed following an initial scoping of the focus area and in consultation with an information specialist. The search strategy combined free text and index terms for the concepts study type (CRTs) and *schools* (box 1). The study type concept was developed based on a sensitive MEDLINE search strategy for identification of CRTs developed by Taljaard *et al.*[28] Cluster design-related terms, 'cluster*',

---

**Box 1    Search strategy implemented in this systematic review**

**Search strategy**
Terms for Randomised Controlled trials:
1. random:.mp.
2. trial. ab,kw, ti.
Cluster design-related terms:
3. "cluster*".ab, kw, ti.
4. "group*".ab, kw, ti.
5. "communit*".ab, kw, ti.
6. 3 OR 4 OR 5
School MeSH term:
7. exp Schools/
Highest precision:
8. 1 AND 2 AND 6 AND 7
9. 8 limited to English language

---

'group*' and 'communit*' were combined with the terms 'random' and 'trial', along with 'Schools'. The search was then limited to English language as our resources make it unfeasible to translate papers (online supplemental table S1).

### Eligibility criteria

Eligible papers will be those reporting the results from school-based CRTs of health-related interventions in the UK for which there is a primary health outcome (including physical and mental health outcomes, health attitudes and well-being) measured on pupils.

The review will include participants who are children of school age in education in the UK. Participants are pupils in preschool, primary school and secondary school settings. The types of eligible clusters include schools themselves, year groups, classrooms, teachers or any other relevant school-related unit. Any health-related intervention(s) will be considered. There must be a control/usual care comparison group within the published article. The primary outcome must be health related and measured on pupils.

All types of CRT design are eligible, including parallel group, crossover and stepped wedge trials. Only definitive CRTs will be included. Only studies published in English will be included and we anticipate that all studies carried out in the UK will be published in English. If more than one publication of the primary outcome result for an eligible CRT is identified, a key study report (index paper) will be designated and used for data extraction.

Papers which do not report the main findings (primary outcome) will be excluded along with feasibility/pilot studies, protocol/design articles, process evaluations, economic evaluations/cost-effectiveness studies, statistical analysis plans, commentaries and papers reporting only findings from mediation/mechanism analyses. Studies for which the primary outcome is not health based (eg, education attainment) will be excluded.

## Screening and selection

All potentially eligible studies will undergo a two-stage screening process.

Stage 1: The titles and abstracts of the studies will be retrieved from MEDLINE (Ovid) and downloaded into Endnote (X9).[29] Any duplicate citations will be removed and remaining citations will be dual screened (KP and OU) for eligibility against the inclusion criteria above. Disagreements will be resolved through discussion.

Stage 2: Full-text articles will be obtained for all papers that are potentially eligible following title and abstract screening. The reviewers (KP and OU) will evaluate articles based on the inclusion criteria using a prepiloted coding method. Any discrepancies which cannot be resolved through discussion will be sent to a third reviewer (ZMX) for a decision.

Reviewers will keep a record of any studies excluded at each step. Results will be reported using a PRISMA flow diagram.

## Risk of bias (quality) assessment

A risk of bias assessment is not necessary for this methodological review as we are not interested in the specific estimates of intervention effect in the included studies. We seek only to describe characteristics of the studies. Some of the information extracted from the papers is indicative of quality in CRTs and this will be summarised as part of our review.

## Data extraction

A data extraction form will be piloted on a random sample of 10 included papers. Any modifications to the form will be made following the pilot. KP will extract data from all eligible papers, while OU will check extraction. Any uncertainty will be resolved by consulting a third reviewer (ZMX). Information will be recorded using a data extraction form in Microsoft Excel.

The following data will be extracted from included articles: characteristics of the participating schools and pupils; intervention type and mode of delivery; health condition/aspect targeted by the intervention; justification for using cluster trial design; unit of randomisation (ie, type of cluster); school-level (or other cluster-level) characteristics used to balance the randomisation; allocation ratio; length of follow-up; number of follow-ups; target sample size (ie, number of schools and pupils); assumptions underlying sample size (eg, ICC, anticipated loss to follow-up); committee that provided ethical approval; activities covered by the consent agreements; primary outcome; reporter of primary outcome (eg, teacher, parent, pupil); method of data collection; achieved sample size; number of schools (clusters) and pupils that were lost to follow-up; analysis method used to estimate intervention effect; baseline factors that were adjusted for in the analysis; value of the ICC in the primary analysis model; methodological challenges that were highlighted by the authors.

Missing information that is not available in the included papers will be obtained from corresponding protocol papers and other sibling publications for the studies. Authors may be contacted for missing or incomplete information and given 2 weeks to respond.

## Data analysis

No formal sample size in terms of the number of required eligible papers has been calculated because we are seeking to obtain all school-based CRTs in the UK to date published in MEDLINE (Ovid). Meta-analysis is not appropriate as the review is focused on summarising methodological characteristics. Study characteristics will be summarised using means and standard deviations (or medians and IQRs) for continuous variables, and numbers and percentages for categorical variables. Appropriate graphs (eg, histograms, line graphs, scatterplots) will also be used to summarise specific features of the data. Challenges reported by authors will be summarised narratively.[30] Statistical analysis will be performed using Stata V.16.[31]

## Patient and public involvement

There has been no contribution from patients or the public to the design of this systematic review protocol.

## DISCUSSION

To our knowledge, this is the first systematic review to describe the characteristics and methodological practices of school-based CRTs of health interventions in the UK. We have a defined search strategy that is tailored to identifying school-based CRTs, selection criteria and a prepiloted extraction strategy. Pilot testing, and subsequent screening and data extraction will be conducted by two independent reviewers, with disagreements resolved by consulting a third reviewer. In doing this we hope to minimise inaccuracy. Additionally, the review aims to cover a range of CRTs conducted in schools for a variety of different health conditions/areas.

Identifying CRTs is challenging as many papers do not explicitly use the word 'cluster' in the title or abstract. We have included terms in our search such as 'group' randomised and 'community' randomised to try and improve the sensitivity, thus widening the search so not to miss any eligible papers. We have also used the exploded Medical Subject Headings (MeSH) term, 'exp School/', in the hope of identifying publications that may state schools or classes as their unit of randomisation.

This review will summarise data using descriptive statistics. Meta-analysis is not used here to pool ICC estimates. Our initial scoping of the literature indicated that most papers do not report the SE of the ICC, which is required for pooling the estimates. Furthermore, the studies to be included in the review will be methodologically and clinically diverse (eg, different outcomes and health conditions). There is, therefore, no true single underlying ICC; rather there is a range of true ICCs specific to different

scenarios. A single pooled ICC from a meta-analysis would not be meaningful and would obscure nuances about how its size depends on the context of the study. It is more useful to summarise the variability in the estimated ICCs as this provides a range of values within which to assess the sensitivity of the sample size calculation to uncertainty about the true value of the ICC in a given scenario.[32]

We have conducted extensive scoping searches in order to best identify the studies of interest. A limitation of the review is that we will limit our search to the MEDLINE (Ovid) database, thus, potentially missing out on articles published in other journals (eg, mental health interventions published in PsycINFO). MEDLINE was used because our research question is to describe the characteristics of trials that evaluate the impact of health interventions on health outcomes. The database includes journals of interest for both physical health and mental health. We have also chosen not to examine grey literature therefore potentially missing out on studies with greater methodological challenges. Also there will be no forward and backward citation searching, but we do have a clearly defined population of papers. Feasibility studies have been excluded, but there are different learning issues from such studies that will be the subject of a separate review. These decisions have been taken to enable the review to be more focused and time-effective.

The review is focused on health-based CRTs in schools. There is a wider literature of other types of intervention (particularly in educational research) that have been evaluated in this setting using the CRT design, but, given the limited resources and the large number of potentially eligible studies identified during scoping, it was considered more relevant and efficient to restrict the review to studies in the health area.[33]

Another limitation of this review is the difficulty in identifying CRTs as many papers do not explicitly use the word 'cluster' in the title or abstract. Therefore, we have included terms in our search such as 'group' and 'community' to try and improve the sensitivity, thus widening the search so not to miss any eligible papers. We have also used the exploded MeSH term, 'exp School/', in the hope of identifying publications that may state schools or classes as their unit of randomisation.

A pragmatic decision has been made to focus on UK studies in order to align with available resources and create a relevant and focused review. There is the possibility that we are missing out on the opportunity to learn from studies in countries that have a similar education system to the UK. Our scoping searches established that there is a considerably large number of eligible papers and we restricted the study eligibility to the UK. Ideally, we would include papers globally, but this is not practical. Despite being focused on the UK, the findings of this review will be of wider interest as many methodological challenges in the design of CRTs will be similar across some countries.

Because of the amount of time children spend in school, it provides a natural setting in which interventions for preventing health problems and improving health outcomes in children can be delivered and evaluated.[13] Cluster randomised controlled trials in the school-based setting are particularly relevant for non-pharmacological interventions, such as social programmes aimed at improving public health[13] and the use of this study design is increasing.[12] Through summarising the methodological aspects of health-related cluster randomised controlled trials conducted in a schools, this review will provide methodology-related knowledge specific to these trials which will help researchers plan future similar studies effectively in the UK and elsewhere.

## ETHICS AND DISSEMINATION

This methodological systematic review does not require ethical approval as only secondary data extracted from papers will be analysed and the data are not linked to individual participants. After completion of the systematic review, the data will be analysed, and the findings disseminated through peer-reviewed publications and scientific meetings.

**Acknowledgements** KP was supported by a PhD studentship funded by the National Institute for Health Research Applied Research Collaboration South West Peninsula (NIHR ARC South West Peninsula). OU was supported by the NIHR ARC South West Peninsula.

**Contributors** KP and OU conceptualised the study. KP drafted the manuscript and incorporated comments from authors for successive drafts. MPN, ZMX and TF contributed to the design and content. All authors read and approved the final manuscript. KP is the guarantor of the review.

**Funding** This work was supported by the National Institute for Health Research (NIHR) Applied Research Collaboration South West Peninsula (NIHR ARC South West Peninsula). Our funded group National Institute for Health Research Applied Research Collaboration South West Peninsula does not have a grant number because we are part of the NIHR infrastructure.

**Disclaimer** The views expressed are those of the author(s) and not necessarily those of the NHS, the NIHR or the Department of Health and Social Care.

**Competing interests** None declared.

**Patient consent for publication** Not required.

**Provenance and peer review** Not commissioned; externally peer reviewed.

**ORCID iDs**
Kitty Parker http://orcid.org/0000-0003-2319-8227
Obioha C Ukoumunne http://orcid.org/0000-0002-0551-9157

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
