## [Reviewer comments · BMJ Open]

ARTICLE DETAILS

TITLE (PROVISIONAL)	Characteristics and practices of school-based cluster randomised controlled trials for improving health outcomes in pupils in the UK: a systematic review protocol
AUTHORS	Parker, Kitty; Nunns, Michael; Xiao, ZhiMin; Ford, Tamsin; Ukoumunne, Obioha

VERSION 1 – REVIEW

REVIEWER	Jan R. Boehnke University of Dundee, UK I am member of a Data Management and Ethics Board related to a Wellcome Trust Strategic Award (MYRIAD) in which Dr. Okoumunne serves as the chief statistician. I am lead methodologist on an Education Endowment Foundation- and NIHR-funded study (NIHR, 14/52/38) where Dr. Ford is the chair of the Advisory Board.
REVIEW RETURNED	26-Oct-2020

GENERAL COMMENTS	The submission "Characteristics and practices of school-based cluster randomised controlled trials for improving health outcomes in pupils in the UK" provides a protocol for a systematic review exploring the characteristics and methodological practices of CRTs delivered in schools to evaluate interventions for improving health outcomes in pupils in the UK. The protocol is well-written and contains element that are in line with the registration on the 7th Aug 2020. One of my key questions was why the review focuses only on the UK, but this is directly answered in the highlights section of the paper. I have only very few comments which mainly relate to some inconsistencies between parts of the stated intent of the review, data extraction and analysis. The main aspect that does not become clear to me from either the PROSPERO registration or the paper itself is the main target purpose of the systematic review. a) In multiple places this is characterised "as the review is focused on summarising methodological characteristics". This is also implied by the title of the submission and the abstract. b) But in contrast to this, the introduction of the submission is largely focused on the importance and measurement of ICCs. And these are also the target identified in the protocol as the main outcome (instead of the study characteristics and features as would have been required in line with the goal stated above). c) The protocol states regarding the strategy for analysis (see also "data analysis" section in the current submission): "Meta-analysis is not appropriate as the review is focused on summarising
--

	methodological characteristics" As the main outcome parameters are ICCs and the percentage of clusters (schools) that dropped-out or were withdrawn, this is unclear as these outcomes are very difficult to summarise narratively, but are actually well-suited for a meta-analysis. d) With view to the introduction I would have expected some overview explaining the characteristics and methodological practices of CRTs and particularly the connected ethical issues (as highlighted by the authors in the abstract). Compared to information on ICCs there is very little information provided on these more general aspects that are mentioned as the main focus of the systematic review. As PROSPERO registrations are fairly brief, it seems the full protocol could offer a clarifying narrative with view to these points. DATABASE: a) Abstract: "indexed in Ovid (MEDLINE)" It seems MEDLINE is the database and Ovid the provider and this should be switched (as in the PROSPERO registration). b) The protocol should provide a brief rationale as to why only this data base is used. DATA STATEMENT: The authors should clarify whether the extracted data will be made fully available or not.
--	---

REVIEWER	Sally Brinkman Telethon Kids Institute, University of Western Australia, Australia
REVIEW RETURNED	09-Nov-2020

GENERAL COMMENTS	The limitation of this review to studies undertaken in the UK significantly reduces the merit of the paper. Expanding to at least, Canada, Australia and other similar such high income country education systems would increase the relevance of the paper to a wider audience. Although the paper is well written, how the work planned will further the scientific literature is not convincing. The paper fails to recognize that many CRTs are undertaken in schools for the purpose of evaluating education programs. Indeed one could argue that the field of education would have a significant contribution to this work, and databases such as ERIC should be considered. The idea that the methodological considerations of CRTs in schools are specific to health interventions is too limited. The knowledge gain for BMJ Open readers is low. In conclusion, as it stands, I am not convinced that this protocol reaches the level of scientific contribution to merit publication.
--

VERSION 1 – AUTHOR RESPONSE

Reviewer' Comments to Authors:

Reviewer 1

The protocol is well-written and contains element that are in line with the registration on the 7th Aug 2020.

AUTHOR RESPONSE: Thank you.

1. The main aspect that does not become clear to me from either the PROSPERO registration or the paper itself is the main target purpose of the systematic review.

a) In multiple places this is characterised "as the review is focused on summarising methodological characteristics". This is also implied by the title of the submission and the abstract.

b) But in contrast to this, the introduction of the submission is largely focused on the importance and measurement of ICCs. And these are also the target identified in the protocol as the main outcome (instead of the study characteristics and features as would have been required in line with the goal stated above).

AUTHOR RESPONSE: The forms for PROSPERO necessitated that we select "main outcomes" for the registration process but, as indicated by our protocol submission, this review aims to summarise a range of methodological characteristics that are of importance for the planning, conduct and analysis of CRTs in schools.

c) The protocol states regarding the strategy for analysis (see also "data analysis" section in the current submission): "Meta-analysis is not appropriate as the review is focused on summarising methodological characteristics" As the main outcome parameters are ICCs and the percentage of clusters (schools) that dropped-out or were withdrawn, this is unclear as these outcomes are very difficult to summarise narratively, but are actually well-suited for a meta-analysis.

AUTHOR RESPONSE: In order to conduct a meta-analysis we would need a point estimate of the ICC and the standard error of (or the 95 confidence interval for) the ICC. Scoping reviews and our knowledge of the literature indicate that the standard error is rarely reported in published reports of cluster randomised trials, so there would be very few ICCs that we could pool in a meta-analysis. There are, however, more substantial reasons why meta-analysis of the ICC would not be appropriate. The studies to be included in the review will be methodologically and clinically diverse (e.g., different outcomes and health conditions). There is, therefore, no true single underlying ICC, rather there is a range of true ICCs specific to different scenarios. A single pooled ICC from a meta-analysis would not be meaningful and would obscure nuances about differences in the ICC across settings. What is more useful is to summarise the variability in the estimated ICC as this provides a range of values within which to assess the sensitivity of the sample size calculation to uncertainty in the true value of the ICC in a given scenario. We will also explore how the ICC differs across key categories of interest (e.g., primary versus secondary school; mental health versus physical health outcomes, etc.). We have added some sentences around this issue in the Discussion section (lines 313 – 323).

d) With view to the introduction I would have expected some overview explaining the characteristics and methodological practices of CRTs and particularly the connected ethical issues (as highlighted by the authors in the abstract). Compared to information on ICCs there is very little information provided

on these more general aspects that are mentioned as the main focus of the systematic review. As PROSPERO registrations are fairly brief, it seems the full protocol could offer a clarifying narrative with view to these points.

AUTHOR RESPONSE: We agree that there are other issues besides clustering that are key to the design of cluster randomised trials, such as ethical procedures, consent and retention. One of the reasons why we are doing the systematic review is to establish how these challenges manifest themselves in the context of school-based studies. We have added some sentences in the Introduction section to highlight other issues (lines 164 – 171).

DATABASE:

a) Abstract: "indexed in Ovid (MEDLINE)" It seems MEDLINE is the database and Ovid the provider and this should be switched (as in the PROSPERO registration).

AUTHOR RESPONSE: Thank you for drawing our attention to this. We have made this change throughout the text.

b) The protocol should provide a brief rationale as to why only this data base is used.

AUTHOR RESPONSE: We chose to search MEDLINE because our research question is to describe the characteristics of trials that evaluate the impact of health interventions on health outcomes. The database contains journals of relevance for both physical health and mental health outcomes. Extensive scoping revealed that the size of the literature is considerable and a pragmatic decision was made to examine MEDLINE exclusively in order to align with available resources, but we do not anticipate that we have missed a salient number of relevant papers as a result of this. We now acknowledge this limitation more clearly in the Discussion section (lines 324 – 335).

DATA STATEMENT:

a) The authors should clarify whether the extracted data will be made fully available or not.

AUTHOR RESPONSE: This paper is part of a wider programme of research to inform the design, conduct and analysis of school-based cluster randomised trials. The extracted data may be made available on request when this work is completed.

Reviewer: 2

1. The limitation of this review to studies undertaken in the UK significantly reduces the merit of the paper. Expanding to at least, Canada, Australia and other similar such high income country education systems would increase the relevance of the paper to a wider audience.

AUTHOR RESPONSE: Our scoping searches established that there is a considerably large number of eligible papers and a pragmatic decision was made to restrict study eligibility to the UK in line with available resources. Ideally, we would include papers globally, including low income countries as well as high income countries, but this is not practical. The choice was between a review that covers a wide range of countries but extracts less information, and a review that covers one country and extracts richer data. There are strengths and limitations either way, but we think the latter option of focussing the review on the UK is the best approach. By eliminating variability between countries we can focus on a considerable number of factors that define variability across schools within a single

school system. Other work undertaken as part of this programme of research will incorporate knowledge from other countries, and we do not underestimate the importance of this.

Despite being focussed on the UK the findings of this review will be of global interest. As the reviewer notes, other high income countries like Australia have a similar school system to the UK, and many of our findings may have applicability to those settings. Two of the co-authors for this review are investigators on a UK-based study for which the sample size calculation was based on estimates of the intra-cluster (intra-school) correlation coefficient from two studies conducted in Australia and Canada, respectively, so we have practical experience of the value of using data from other countries.¹ Many methodological challenges in the design of cluster randomised trials will be similar across different settings. We have added some text to the Discussion section to more clearly highlight this issue (line 337 – 314).

2. Although the paper is well written, how the work planned will further the scientific literature is not convincing. The paper fails to recognize that many CRTs are undertaken in schools for the purpose of evaluating education programs. Indeed one could argue that the field of education would have a significant contribution to this work, and databases such as ERIC should be considered. The idea that the methodological considerations of CRTs in schools are specific to health interventions is too limited.

AUTHOR RESPONSE: We agree with the reviewer that there is a considerable wealth of knowledge to be obtained from the way cluster randomised trials have been conducted in educational research and acknowledge that the design has a longer history in the context of education research than it does in health research. We reiterate that this review seeks to summarise studies that have evaluated the effect of health interventions on health outcomes. Given the limited resources and the large number of potentially eligible studies identified during scoping, it was considered more relevant and efficient to focus on studies in the health area. For the same reason we focus on using the MEDLINE database rather than ERIC. Again, our wider programme of research will incorporate learning from the education research field. We now provide comment on this issue in the Discussion section (lines 352 – 356).

3. The knowledge gain for BMJ Open readers is low. In conclusion, as it stands, I am not convinced that this protocol reaches the level of scientific contribution to merit publication.

AUTHOR RESPONSE: BMJ Open is read by a diverse range of interdisciplinary researchers with varying levels of practical knowledge of designing cluster randomised trials. As researchers who are actively involved in school-based cluster randomised trials we know there is a need for this review and that the findings will be of great interest to the research community, including those with expertise in delivering CRTs, and wider stakeholders (e.g., teachers, policy makers). This will be borne out by the level of interest in the resulting findings.

From our scoping activity, we have found a wealth of health-focused school-based cluster randomised trials, but despite this abundance of research being conducted in this area, there has been no methodological systematic review to date. We can see the merit of this review in relation to similar systematic reviews that have been published in clinic and primary care settings. This review represents leading-edge research in an expanding field and will provide a point of reference to researchers in this area and will spearhead a series of publications on this topic.

References

1. Kuyken W, Nuthall E, Byford S, et al. The effectiveness and cost-effectiveness of a mindfulness training programme in schools compared with normal school provision (MYRIAD): study protocol for a randomised controlled trial. *Trials* 2017;18(1):194. doi: 10.1186/s13063-017-1917-4

VERSION 2 – REVIEW

REVIEWER	Jan R. Boehnke University of Dundee, UK As indicated in my first review, I am the member of a the Data Management and Ethics Committee for a programme of studies for which one of the authors is the lead statistician.
REVIEW RETURNED	16-Dec-2020

GENERAL COMMENTS	Thank you for inviting me to review the revisions for the protocol "Characteristics and practices of school-based cluster randomised controlled trials for improving health outcomes in pupils in the UK". In my opinion the authors have been responsive to the feedback provided and offered the required clarifications. With the additions to the introduction the purpose of the review reads clearer to me and the introduction is now also more balanced with view to aspects other than ICC estimation. The addition as to the difficulties regarding summarising ICCs and the rationale of not doing so is clear. The minor points were addressed as well. Prompted by the other points raised in the peer-review process, I revisited BMJOpen's expectations regarding peer-review or protocols (webpage, 16.12.2020): "Reviewers will be instructed to review for clarity and sufficient detail. The intention of peer review is not to alter the study design. Reviewers will be instructed to check that the study is scientifically credible and ethically sound in its scope and methods, and that there is sufficient detail to instil confidence that the study will be conducted and analysed properly." As to the points required to be checked for review, I can only say that I think that the new version is clear, offers sufficient detail and provides good evidence that the study is credible as a scientific project, it is ethically sound in scope and methods, and there is sufficient detail that the study will be conducted and analysed properly.
--